# Significance of feedforward architectural differences between the ventral visual stream and DenseNet

**Bryan Tripp**
Centre for Theoretical Neuroscience
University of Waterloo, Canada
bptripp@uwaterloo.ca

## Abstract

There are many differences between convolutional networks and the ventral visual streams of primates. For example, standard convolutional networks lack recurrent and lateral connections, cell dynamics, etc. However, their feedforward architectures are somewhat similar to the ventral stream, and warrant a more detailed comparison. A recent study found that the feedforward architecture of the visual cortex could be closely approximated as a convolutional network, but the resulting architecture differed from widely used deep networks in several ways. The same study also found, somewhat surprisingly, that training the ventral stream of this network for object recognition resulted in poor performance. This paper examines the performance of this network in more detail. In particular, I made a number of changes to the ventral-stream-based architecture, to make it more like a DenseNet, and tested performance at each step. I chose DenseNet because it has a high BrainScore, and because it has some cortex-like architectural features such as large in-degrees and long skip connections. Most of the changes (which made the cortex-like network more like DenseNet) improved performance. Further work is needed to better understand these results. One possibility is that details of the ventral-stream architecture may be ill-suited to feedforward computation, simple processing units, and/or backpropagation, which could suggest differences between the way high-performance deep networks and the brain approach core object recognition.

## 1   Introduction

Deep convolutional networks trained for object recognition have a number of things in common with the primate ventral stream, including architectural similarities (e.g. spatially localized connections), comparable performance, and closely related representations [1]. There are also many differences, notably related to cell and network dynamics and development of representations over time. However, feedforward convolutional networks appear to be promising abstract models of core object recognition. Can convolutional networks be used to model the signals and transformations of core object recognition in detail? A recent study developed a data-driven convolutional network that was consistent with many architectural details of primate visual cortex [2]. However, that study reported performance far below state of the art when the ventral stream was trained on CIFAR-10. Here I confirm this result, and investigate whether there is a specific feature of the cortex-like architecture that limits performance. There is not. Rather, multiple architectural details are responsible for the performance gap. This may suggest that the feedforward architecture of the ventral stream is not particularly well suited to core object recognition in the idealized context of standard deep-network training.

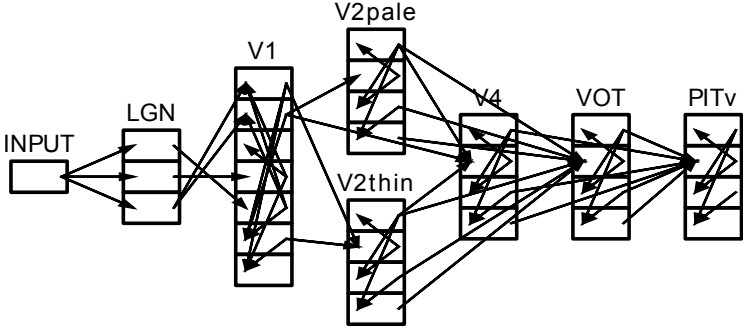

Figure 1: Structure of the ventral-stream-like network. Except where mentioned, parameters were taken from [2]. The boxes within each area indicate sub-populations with distinct connections. For most cortical areas, these are layers L2/3, L4, L5, and L6. For V1, these are L2/3(blob), L2/3(interblob), L4B, L4C$\alpha$, L4C$\beta$. For LGN, they are parvo, magno, and koniocellular divisions.

## 2 Ventral-Stream Network

I experimented with a simplified sub-network of the macaque single-hemisphere visual cortex architecture of [2]. This sub-network spans much of the ventral stream (Figure 1, including V1, V2 (thin and pale stripes), V4, VOT, and PITv. The retrograde-tracer measure, fraction labelled neurons extrinsic to the injection site (FLNe), was used to identify and omit the weakest inter-area connections (FLNe<0.02). To make training tractable, the number of neurons in each population was reduced by a factor of ten. The full visual resolution was taken to be 500x500 pixels, and smaller images from ImageNet and CIFAR-10 were taken to activate the central part of the visual field. L2/3 of PITv was connected to a final convolutional layer with 64 channels, and then either two 512-unit fully-connected layers (for CIFAR-10) or one 2048-unit fully-connected layer (for ImageNet), followed by a softmax classifier layer. Each linear layer in the model was followed by batch norm and ReLU layers. The code for this model can be found at github.com/bptripp/calc.

## 3 Results

[2] reported validation accuracy of 79% on CIFAR-10, for a similar ventral-stream sub-network that omitted connections with FLNe<0.15, and was trained for 50 epochs with the Adam update algorithm. In the present study, networks were trained for 300 epochs, using SGD with momentum 0.9, starting with learning rate 0.1 and reducing it by 10x every 100 epochs. This resulted in validation accuracy of 84.59%. A standard DenseNet (github.com/kuangliu/pytorch-cifar) was trained using the same procedure, resulting in validation accuracy of 95.36%.

To understand the basis of this performance gap, I created hybrid networks, with features of both the ventral-stream network (VSN) and DenseNet. The VSN has a wide range of kernel sizes, optimized to fill realistic receptive field sizes. In the first hybrid (H1), all kernel sizes of the VSN were set to 3x3. The VSN also has a wide range of sparsity, with some connections consisting mostly of zeros. In the second hybrid network (H2), in addition to using 3x3 kernels, I eliminated pixel-wise sparsity, and limited channel-wise sparsity so that at least half of the input channels were used in each connection. Thirdly (H3), I replaced each layer with a two-layer bottleneck module, specifically a 1x1-kernel layer followed by 3x3 layer with four times fewer channels. The number of channels in the second layer was 0.3x the original number of channels. Fourthly (H4), I replaced the LGN layers with a DenseNet-like 24-channel input stage, eliminated remaining sparsity, and organized the remaining layers into three blocks with DenseNet-like transition layers between them. Each block grouped layers with the same spatial resolution. Connections that spanned multiple blocks were replaced by connections with the transition layers. Fifthly (H5), I set the numbers of channels in each module within a given block equal to the block-wise average, emulating the block-wise constant "growth rate" parameter of DenseNet. Finally (H6), I added any missing connections within each block, introducing many connections that do not exist in the brain. These changes are summarized in Table 1.

Table 1: Summary of architectural changes to make VSN more similar to DenseNet.

| Architecture Variation | Short Description |
| --- | --- |
| H1 (3x3 kernels) | Various kernel sizes all replaced with 3x3 |
| H2 (dense kernels) | No pixel-wise sparsity; low channel-wise sparsity |
| H3 (bottlenecks) | Each layer replaced with two-layer bottleneck |
| H4 (single-resolution blocks) | 3 blocks with DenseNet-like transitions |
| H5 (uniform growth rate) | # new channels in each layer consistent within blocks |
| H6 (dense connections) | All possible forward connections within blocks |

Table 2: CIFAR-10 validation accuracy in hybrid ventral-stream/DenseNet architectures (details in text). The left column shows results of cumulative modifications that make the ventral-stream network increasingly similar to a DenseNet. The right column shows the results of each change alone. Most cumulative changes improve performance. Most individual changes do not substantially improve performance, with the exception of making the growth rate uniform. However, this change in isolation substantially increased the number of parameters, and controlling for this eliminated the benefit. The final modification was not performed alone, because the baseline ventral-stream network has no notion of blocks that could contain all possible feedforward connections.

| Architecture Variation | Validation Accuracy | |
| --- | --- | --- |
| Baseline ventral-stream network | 84.59% | |
| Modifications from baseline | (cumulative) | (alone) |
| H1 (3x3 kernels) | 85.33% | 85.33% |
| H2 (dense kernels) | 88.09% | 85.53% |
| H3 (bottlenecks) | 76.79% | 73.41 % |
| H4 (single-resolution blocks) | 93.15% | 77.35 % |
| H5 (uniform growth rate) | 92.5% | 87.11% |
| H6 (dense connections) | 93.89% | - |
| DenseNet | 95.36% | |

Most of these modifications improved performance, when applied cumulatively to make the ventral-stream increasingly similar to a DenseNet (Table 2, left result column). Making the growth rate uniform resulted in a slight performance drop. Replacing simple layers with bottleneck modules sharply impaired performance, including on training data (83% correct), although performance recovered in the next variation with transition layers. The bottleneck experiment was repeated with slight variations (i.e. a different training schedule, growth rate 0.2x rather than 0.3x the original number of channels, and a DenseNet-like input stage), and similar performance impairment was found. With these exceptions, hybrid networks performed better the more they had in common with DenseNet.

Similar modifications were also applied one at a time (Table 2, right result column). Improvement due to dense kernels alone was less pronounced than in conjunction with 3x3 kernels. To approximate DenseNet blocks (H4), most of the connections between layers with different resolutions were eliminated. However, some layers only had incoming connections from higher-resolution layers. For these layers, the single inbound connection with the largest number of parameters was retained. This resulted in an approximately block-wise structure, without introducing connections that do not exist in the brain. This change strongly impaired performance. To approximate DenseNet's uniform growth rate within each block (H5), the number of channels in each layer was changed to the average number of channels among layers with the same resolution. This improved performance, but the redistribution of channels also caused a large increase in the number of parameters in the network. Repeating this experiment while controlling for the number of parameters reduced validation accuracy slightly below baseline (83.71%). Overall, each of these changes alone had little benefit.

As discussed in [2], connection sparsity in the VSN is qualitatively distinct from various forms of sparsity in deep learning. I further explored the effect of sparsity on performance. Each connection in the VSN has two sparsity parameters, $\sigma$ (pixel-wise probability of non-zero weight), and $c$ (channel-wise probability of non-zero weights); the probability that a weight is non-zero is $\sigma c$. I chose non-zero

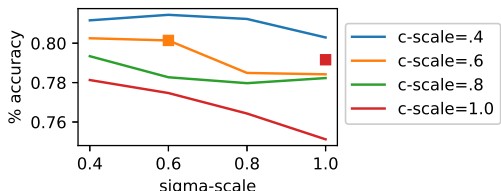

Figure 2: Effect of sparsity parameters $c$ (channel-wise) and $\sigma$ (element-wise) on performance. Different lines show CIFAR-10 performance of different ventral-stream networks with log-$c$ multiplied by various scale factors. Larger c-scale means more sparsity. The horizontal axis indicates log-scaling of $\sigma$. Networks trained for 50 epochs with Adam; learning rate 0.001. The red square shows performance with both scale factors set to 1, but with the number of channels in each layer increased by a constant factor to make the total number of parameters within 1% of the network with both scale factors set to 0.6 (orange square).

channels at random, and non-zero weights using SNIP [3]. The sparsity parameters vary over several orders of magnitude in different connections. I trained variations of the VSN in which I scaled the logarithms of these parameters (e.g. $\sigma = .0001$ with a scale factor of .5 would become .01). Unsurprisingly, reductions in sparsity tended to improve performance (Figure 2). So, physiologically realistic connection sparsity seems to be counter-productive in these deep-network models of core object recognition. A network with full sparsity, but with numbers of channels increased so that the total number of parameters matched that of a network with scale factors of 0.6, performed nearly as well as the network with scale factors of 0.6 (squares in Figure 2. This suggests that sparsity affects performance largely via the number of network parameters.

Representations in DenseNet are closely related to those in primate ventral stream [1]. However, I wondered whether, despite poorer performance of the VSN, it might have even more realistic representations due to its architecture. To explore this issue, I performed approximations of several experiments from the primate literature, with both a DenseNet-161 model pretrained on ImageNet (from torchvision.models), and a VSN trained on ImageNet, with c-scale=$\sigma$-scale=0.5. The VSN was trained with SGD for 90 epochs, with learning rate 0.1, decreasing by 10x every 30 epochs; best validation accuracy 63.14%(top-1)/84.97%(top-5). I recorded from the second-last ReLU layer that followed a convolutional layer in each network, and compared with published recordings from macaque inferotemporal cortex. The two networks had somewhat different representations, but overall these experiments did not reveal either to be much more realistic (Figure 3).

The representation results were sensitive to implementation details such as whether stimulus images were normalized or cropped (center 224x224 pixels taken from 256x256 images). It is not clear whether the stimulus images' standard deviations should be normalized (as is usually done with natural images). This is because they consist, as in the primate experiments, of small content on a larger neutral background. In the size tuning experiments, normalizing each image would give the larger stimuli systematically lower contrast, while in the other experiments, normalizing would give thin objects (e.g. a hammer) higher local contrast. The results in Figure 3 are without normalization or cropping, but other choices produced somewhat different results. Normalization tended to narrow size-tuning bandwidth in both VSN and DenseNet. The RDMs from normalized stimuli had correlations of $r = 0.62$ (VSN) and $r = .68$ (DenseNet) with those in the figure. The RDMs from normalized and cropped stimuli had correlations of $r = 0.34$ (both VSN and DenseNet) with those in the figure. Sensitivity to such factors complicates comparison with macaque data.

## 4 Discussion

Architectural details of the ventral-stream model tended to impair core object recognition in our deep-learning setting. This may indicate inaccuracies in the architecture of [2]. Alternatively, it may suggest that the feedforward ventral-stream architecture is not well suited for such idealized settings, or even that it is not optimized specifically for core object recognition. The ventral stream architecture might be better suited for more diverse or naturalistic tasks, or more biological mechanisms. For example, biological schemes such as predictive coding may use sparse feedforward connections more

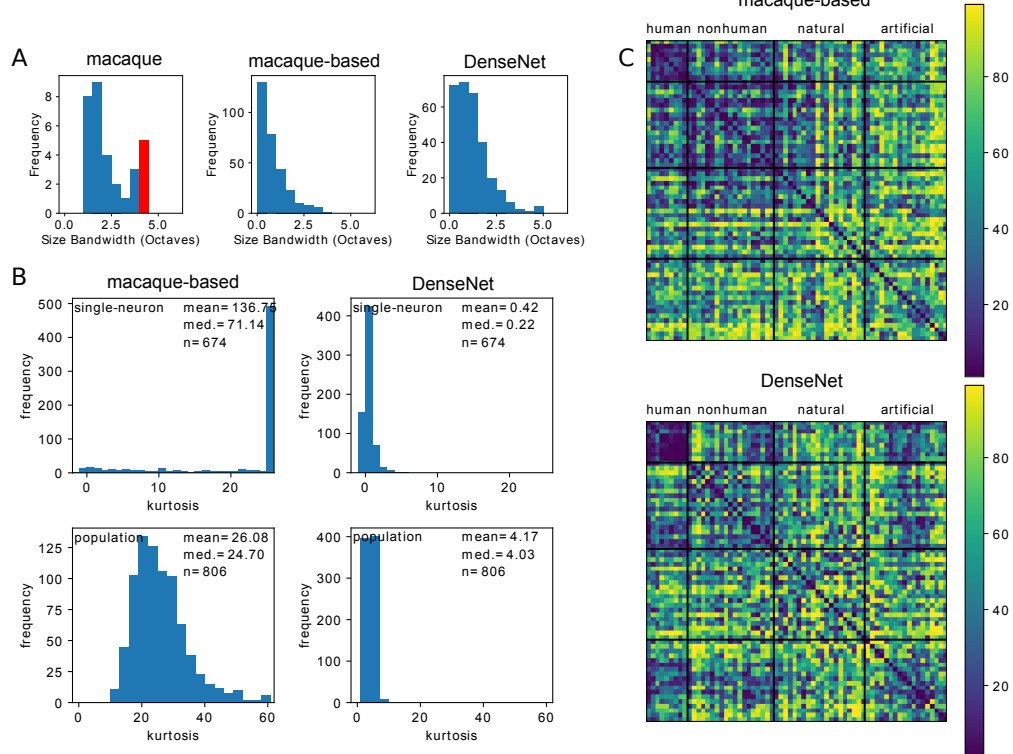

Figure 3: Examination of higher-level representations in the ventral-stream network and DenseNet. A, size-tuning bandwith. Macaque data replotted from [4] (red means $\geq 4$, the limit of the range tested with the monkeys). Both networks seem to have somewhat narrower size tolerances than monkeys. B, Distributions of single-unit selectivity (top) and population sparseness (bottom), with stimuli from [5]. In the monkey inferotemporal cortex data, mean selectivity is 3.5, and mean sparseness is 12.51. The ventral-stream model has much higher means, and DenseNet has much lower means. C. Representational dissimilarity, using a subset of images from [6]. The plotted values are percentiles of one minus the Pearson correlations between responses to different stimuli. Monkey cell data shows relatively low values (high similarity) throughout the lower-right quarter of this matrix (spanning non-animal natural and artificial images) [6], but neither of the deep networks does.

efficiently, or the connection pattern of the ventral stream may be better suited to extracting certain feature combinations in unsupervised learning than to communicating gradients through many layers.

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
