# OpenReview forum: "Significance of feedforward architectural differences between the ventral visual stream and DenseNet"
_NeurIPS.cc/2019/Workshop/Neuro_AI — Real Neurons & Hidden Units @ NeurIPS 2019 Poster_

### Official Review · AnonReviewer3 · 2019-09-24
**interesting direction of unifying bottom-up cortex-like models with top-down large-scale ML models, but needs more quantification**

**Clarity:** 4

**Comment:**

Bridging bottom-up models from neuroscience with top-down models from Machine Learning is a promising direction.
This paper shows some interesting results on which architectural changes need to be made to the bottom-up model, and finds that increased sparsity harms performance.
I would have liked to see
* H1-6 individually to isolate what improves performance by itself/only in combination
* the sparsity analysis in Fig. 2 controlled for network size
* the comparison to neural data quantified, as it is otherwise very hard to make any judgments about which model is better than the other

**Category:**

AI->Neuro

**Clarity Comment:**

Overall, it is clear what steps the authors take.

The following points were confusing to me:
1. The first paragraph of the results describes the different changes to the architecture with a lot of text. This would be easier to digest with at least a list (one item for H1, H2 etc.), or ideally a table that clearly marks what the differences are, starting from the ventral-stream-like network, going over H1 H2 etc., to the DenseNet architecture.
2. line 46 states that there is a readout-head for ImageNet, but the paper only shows CIFAR-10 results

**Evaluation:**

3: Good

**Importance:**

4: Very important

**Importance Comment:**

After the field's initial success of broadly modeling the ventral stream, this study analyzes which tweaks make a data-driven ventral-stream-like network built to resemble architectural details in cortex better at performing on CIFAR-10. There are still major differences between ML models now used to predict brain activity and the actual implementation in cortex and this paper takes first steps to bridge that gap.

**Intersection:**

5: Outstanding

**Intersection Comment:**

The paper compares a neuroscience model (data-driven ventral-stream-like network with cortex-like architectural features) with a Machine Learning model (DenseNet; which has also been shown to predict neural activity and human behavior) and qualitatively evaluates both on macaque data. This general approach is a good direction to combine bottom-up cortex-like architectural features with top-down large-scale neural networks from Machine Learning.
One interesting finding is that physiologically realistic connection sparsity seems to stand in contrast with high task performance. I wonder whether this means we simply need to build bigger networks to accommodate the sparsity or where the mismatch is coming from.

**Rigor Comment:**

The paper starts from a previously published ventral-stream-like network and cumulatively changes its architecture to resemble DenseNet, which is a high-performance ML model with high Brain-Score (i.e. it predicts neural populations + behavior). The cumulative changes are convincing and more and more closely approach DenseNet CIFAR-10 performance.
I would have liked to see all of the changes by themselves instead of only in aggregation to better identify which changes are important or whether it's really the interplay of all those changes that improve performance. Overall though, this analysis is well-done.

Figure 2 tests the effect of sparsity on accuracy and finds that more sparsity harms performance. This is an interesting finding, but I would have liked more controls on the network size: i.e. if you increase the network size, can you still train to remedy the accuracy losses from increased sparsity.

The comparison to macaque neural data in Figure 3 offers a nice fine-grain view at differences of classical measures used in neuroscience (size-tuning bandwidth, single-unit selectivity, population sparseness, and RDMs). However, the results of this analysis are difficult to put into context since model-match-to-brain is not explicitly quantified, but rather only relies on visual comparison. Additionally, the macaque data is only shown for size-tuning, but not for the other three properties. It is thus hard to say which of the models matches the brain more closely.

**Technical Rigor:**

3: Convincing

---

### Official Review · AnonReviewer2 · 2019-09-26
**Sensible, if preliminary, exploration of a continuum of architectures from biological to task-optimised**

**Clarity:** 4

**Comment:**

-- A nice idea, to systematically explore the space between biology-optimised and computer-vision-optimised DCNN architectures.

-- There seem to be some simple ways to improve the set of experiments comparing the networks to macaque/human data. For example, by calculating the representational dissimilarity matrix correlations for each of the intermediate architectures to human and macaque data from Kriegeskorte et al. (2008)'s Neuron paper.

-- The conclusion that the ventral stream may not be "optimised specifically for core object recognition" seems like a bit of a leap. At most, the poor recognition performance suggests the ventral stream may not be optimised specifically for Imagenet- or CIFAR-like tasks in which one must name the main object in decontextualised static images. Put this way, it seems almost certain that mammalian ventral stream is *not* optimised specifically for this task. A more naturalistic definition of "core object recognition" might be recognising the identity and properties and affordances of objects in dynamic visual input, which the ventral stream likely is optimised for.

**Category:**

Neuro->AI

**Clarity Comment:**

-- Generally clearly written.

-- Figures could be more clearly labelled. E.g. a title in Figure 2 indicating that these results concern sparsity. In Figure 3A, an indication of what the red bars distinguish (I still don't fully understand even from the caption - "red means >= 4"....but then this same convention is not applied to the DenseNet histogram?). In Figure 3B, does the red vs blue colour coding indicate anything, or just distinguish single-neuron from population plots? If the latter, these would be better all blue, as the current colour scheme suggests some correspondence with the red vs blue bars of the size-tuning plots in 3A.

-- Representational Similarity Analysis in Figure 3C is very unclear. Figure caption describes this as a representational *similarity* matrix, but then says that macaque data show "low values" for inanimate vs inanimate pairwise entries, implying that the matrix actually shows *dissimilarity* values (but which metric?). Caption should be clarified and figure should include a colour map legend indicating what distance measure is used.

**Evaluation:**

3: Good

**Importance:**

4: Very important

**Importance Comment:**

-- Interesting and timely to explore the architectural design space between models constrained to match biological vision, and those constrained to perform static-image object-classification well. No major insights yet from these particular results, but it is helpful to see that the performance gulf is likely due to many small features, rather than any on single architectural difference.

**Intersection:**

5: Outstanding

**Intersection Comment:**

-- Strong combination of neuroscience and machine learning. Constructs a continuum of models stretching from maximally-biologically-informed, to engineer-optimised, to try to resolve a discrepancy in performance between the two approaches to network architecture choice.

**Rigor Comment:**

-- Architectural and training details described in reasonable detail given the available space.

-- The second set of experiments, testing how well the various architectures match primate brain data, seem rather cursory. Given that the motivation given for choosing Densenet as a target architecture was its relatively high BrainScore, it is odd that BrainScore (or any of its subcomponent scores) is not calculated for any of the intermediate architectures. The only things presented towards this are qualitative histograms of size-tuning and sparsity in the two endpoint architectures (VSN and Densenet). Representational dissimilarity matrices are also shown, for these two architectures only, with no quantification of how well either of these predicts macaque or human matrices.

**Technical Rigor:**

3: Convincing

---

### Official Review · AnonReviewer1 · 2019-09-26
**novel investigation of various network architectures in a "biologically-realistic network"**

**Clarity:** 4

**Category:**

Common question to both AI & Neuro

**Clarity Comment:**

I think more explanation of the VSN would have been helpful.

**Evaluation:**

4: Very good

**Importance:**

4: Very important

**Importance Comment:**

The authors investigate various architecture manipulations of a "ventral-stream-like network" and its effect on object recognition performance. Despite the more "biological" architecture, the VSN was not more similar to IT responses. A comparison to V4 would have been useful and increased the importance.

**Intersection:**

4: High

**Intersection Comment:**

Explores what sorts of manipulations hurt or help object recognition performance.

**Rigor Comment:**

Rigorous study. Performance is greatly improved by adding dense connections. Is this a consequence of the increase in the number of free parameters?

**Technical Rigor:**

4: Very convincing

---

### Decision · Program_Chairs · 2019-10-02

Accept (Poster)